# Metabolic Profiling and Stable Isotope Analysis of Wines: Pilot Study for Cross-Border Authentication

**DOI:** 10.3390/foods13213372

**Published:** 2024-10-23

**Authors:** Marius Gheorghe Miricioiu, Roxana Elena Ionete, Diana Costinel, Svetlana Simova, Dessislava Gerginova, Oana Romina Botoran

**Affiliations:** 1ICSI Analytics Group, National Research and Development Institute of Cryogenic and Isotopic Technologies—ICSI Rm. Vâlcea, 4 Uzinei Street, 240050 Râmnicu Vâlcea, Romania; marius.miricioiu@icsi.ro (M.G.M.); roxana.ionete@icsi.ro (R.E.I.); diana.costinel@icsi.ro (D.C.); 2Bulgarian NMR Centre, Institute of Organic Chemistry with Centre of Phytochemistry, Bulgarian Academy of Sciences, “Acad G. Bonchev” Street, Bl. 9, 1113 Sofia, Bulgaria; svetlana.simova@orgchm.bas.bg (S.S.); dessislava.gerginova@orgchm.bas.bg (D.G.)

**Keywords:** wine authenticity, metabolic profile, stable isotopes, geographical origin differentiation

## Abstract

Globalization and free market dynamics have significantly impacted state economies, particularly in the wine industry. These forces have introduced greater diversity in wine products but have also heightened the risk of food fraud, especially in high-value commodities like wine. Due to its market value and the premium placed on quality, wine is frequently subject to adulteration. This issue is often addressed through regulatory trademarks on wine labels, such as Protected Designation of Origin (PDO) and Protected Geographic Indication (PGI). In this context, the metabolic profiles (organic acids, carbohydrates, and phenols) and stable isotope signatures (δ^13^C, δ^18^O, D/H_I_, and D/H_II_) of red and white wines from four agroclimatically similar regions were examined. The study explored how factors such as grape variety, harvest year, and geographical origin affect wine composition, with a particular focus on distinguishing samples from cross-border areas. Multivariate statistical analysis was used to assess the variability in wine composition and to identify distinct groups of samples. Preliminary results revealed that organic acids and volatile compounds were found in lower concentrations than carbohydrates but were significantly higher than phenols, with levels ranging between 1617 mg/L and 6258 mg/L. Carbohydrate content in the wines varied from 8285 mg/L to 14662 mg/L. Principal Component Analysis (PCA) indicated certain separation trends based on the variance in carbohydrates (e.g., fructose, glucose, galactose) and isotopic composition. However, Discriminant Analysis (DA) provided clear distinctions based on harvest year, variety, and geographical origin.

## 1. Introduction

The global economic impact of wine production and commerce is significant, far surpassing other commodities and drawing considerable attention from authorities worldwide. According to statistics compiled by the International Organization of Vine and Wine (OIV), global wine production in 2022 reached 262.1 million hectoliters, with a corresponding production of 77.3 million tons of fresh grapes. This represents a decrease from the 276.9 million hectoliters of wine produced in 2000, despite an increase in grape production from 64.1 million tons in the same year. This trend suggests a growing market demand for higher-quality wine, as the industry shifts focus toward producing more refined and valued products (https://www.oiv.int/what-we-do/global-report?oiv, accessed on 15 June 2024).

Wine production involves substantial costs and demands extensive expertise, as it is regarded as a luxury commodity in many markets, particularly in the globalized economy. Wine quality is heavily influenced by factors such as terroir, geographic origin, grape variety, and production methods. These elements are frequently emphasized on wine labels through trademarks like Protected Designation of Origin (PDO) and Protected Geographic Indication (PGI), which serve to indicate the product’s premium status and quality [1,2,3,4]. A consumer’s first interaction with a wine bottle is visual, primarily through its label. However, unlike other products, the quality of wine can only be fully assessed once it is tasted, making the label crucial in shaping consumer expectations. Unfortunately, wine counterfeiting is a widespread issue. Common fraudulent practices include misrepresenting geographical origins, selling wines without geographical protection as if they were from renowned regions. Other forms of fraud involve mislabeling grape varieties, where wines made from blends (especially reds like Cabernet Sauvignon or Merlot) are falsely marketed as single-variety wines. Additionally, there is misrepresentation of the vintage year and dilution of wine with water or lower-quality products [5,6,7,8,9]. These fraudulent practices not only deceive consumers but also alter the chemical composition and sensory characteristic of the wine, leading to significant economic gain for the perpetrators at the expense of consumer trust. This phenomenon has become increasingly common and poses a serious issue for both consumers and the legitimate wine industry [10].

Consequently, wine adulteration and undisclosed origins have become increasing concerns, impacting both consumer health and the market through unfair competition, leading to economic losses for traditional wine-producing countries. To address these challenges, various analytical methods have been employed, including isotope-ratio mass spectrometry (IRMS), nuclear magnetic resonance (NMR) for metabolomic profiling, SNIF-NMR for determining deuterium distribution in ethanol, inductively coupled plasma mass spectrometry (ICP-MS) for elemental analysis, ultra-high-performance liquid chromatography (UHPLC) for polyphenol composition, and gas chromatography coupled with mass spectrometry (GC-MS) to identify aroma components. These methods contribute, to varying degrees, to detecting unconventional winemaking practices or determining the wine’s origin [2,11,12,13,14]. Among these methods, the International Wine Office and the European Commission have officially endorsed SNIF-NMR and IRMS to monitor chaptalization, the practice of adding sugar to fermenting grape must to boost the final alcohol content (https://www.eurofins.com/food-and-feed-testing/food-testing-services/authenticity/snif-nmr-concept/, accessed on 15 June 2024). SNIF-NMR is effective because the distribution of deuterium in sugars differs between plants using different photosynthetic pathways (C3, C4, CAM), and the hydrogen isotope ratio (D/H)_I_ of the methyl group in ethanol is indicative of the sugar’s botanical origin. Also, the deuterium abundance in water from the environment where the plant was grown varies by region, influencing the hydrogen (D/H)_II_ of the methylene group of ethanol, which can provide potential information about its region’s climatic conditions [9,11]. Additionally, stable isotope analysis, particularly focusing on carbon-13 (δ^13^C) and oxygen-18 (δ^18^O), plays a crucial role in the authentication and geographic origin verification of wines [15,16,17,18,19,20]. The isotopic composition of biogenic materials, including wine, is significantly influenced by environmental conditions, which vary regionally and are reflected in the isotope ratios of the wine’s components. As with the D/H ratio, the δ^13^C and δ^18^O isotopic ratios in wine are influenced by several climatic and environmental factors. For instance, δ^13^C is positively correlated with temperature and negatively with precipitation during grape ripening. Similarly, δ^18^O values in wine water are affected by regional climatic parameters such as temperature, precipitation, humidity, and evapotranspiration. These factors contribute to region-specific δ^18^O values, making it possible to discriminate between wines from different geographic areas. Furthermore, the δ^18^O ratio is sensitive to altitude and can effectively differentiate wines from various subregions [21]. Also, the homonuclear and heteronuclear 2D NMR spectroscopy have a great implication in wine compound identification for further product authentication and quality control [22]. Furthermore, NMR-based metabolomics was used for wine differentiation according to their terroir, vintage, and variety [23,24,25,26]. There were found strong correlations between metabolites and climatic conditions, clay, and soil organic matter content [27].

Differentiating wines from vineyards located in cross-border regions is challenging due to the often similar environmental conditions in these areas [28,29]. Therefore, it is essential to develop reliable methods for verifying the declared geographic origin of wines. This may involve the use of emerging complementary techniques and chemometric analysis, which can also contribute to establishing databases for more precise identification and verification of geographic origin.

This study aims to assess the effectiveness of two analytical techniques—^1^H-NMR and IRMS (stable isotopes)—combined with multivariate statistical analysis, in overcoming the challenges posed by similar climatic conditions. The objective is to differentiate wines from four geographically close agroclimatic regions in the Drăgășani area of Romania, a region with a rich winemaking tradition, based on geographic origin, grape variety, and harvest year. This differentiation is essential for protecting certified, high-value wines from being substituted with cheaper, adulterated, or mislabeled products. Ensuring wine authenticity is crucial for meeting the expectations of both the industry and consumers, and can also contribute to establishing a wine as a symbol of national pride.

## 2. Materials and Methods

### 2.1. Sample Collection

Samples of eight different red and white wine varieties, including both international and local types—such as Merlot, Fetească Neagră, Sauvignon Blanc, Cabernet Sauvignon, Crâmpoșie, Negru de Drăgășani, Chardonnay, and Fetească Regală—were collected to differentiate them based on geographic origin, grape variety, and vintage. These wines were sourced directly from reputable producers, in their original bottles with labels indicating Protected Designation of Origin (PDO). The samples came from four distinct agroclimatic regions: Sâmburești, Drăgășani, Dobrușa, and Spârleni (Appendix A), and were produced in various years (2019, 2020, 2021, and 2022). After collection, the samples were coded and stored in a refrigerator until chemical analysis. Detailed information about the selected study sites, grape varieties, and vintages is provided in Table 1.

### 2.2. ^1^H NMR Metabolomic Profile

The protocol applied for the metabolite extraction for NMR investigation was based on different literature studies [30,31,32,33]. Earlier, a buffer solution was obtained by adding 2.75 g potassium phosphate (KH_2_PO_4_), 3.9 g sodium azide (NaN_3_, extra pure), 20 mg 3-(trimethylsilyl) tetradeutero propionic acid sodium salt (98%, TSP) internal standard, and deuterium oxide (D_2_O), up to 20 g of mixture. The prepared buffer solution was adjusted to a pH of 3.1 by using HCl (1N) and NaOH (1N). The internal standard has two roles: to calibrate the axes of the ^1^H-NMR spectrum and to quantify the metabolite concentration according to its signal by applying the equation from a literature study which was validated for quantitation of compounds in wine [34].
$$m_{x}=\frac{M_{x}}{M_{std}}\times \frac{n_{std}}{n_{x}}\times \frac{A_{x}}{A_{std}}\times m_{std}$$
where *m_x_* and *m_std_* are the masses of the analyte and the standard (g), *M_x_* and *M_std_* are the molecular weights of the analyte and the standard (g/mol), *n_x_* and *n_std_* are the numbers of protons of the analyte signals and standard, *A_x_* and *A_std_* are the areas for the selected signals of the analyte and standard.

Thus, the wine samples were passed through PTFE membrane filters (0.45 µm, 25 mm diameter, Millipore, Burlington, MA, USA) and then 0.444 mL buffer solution was added to each 4 mL of filtered solution. The pH was adjusted to 3.1 by using the BTpH titration unit (Bruker France SAS, Wissembourg, France), in order to minimize the variation of the signal positions between the spectra [31]. All the samples were transferred in 5 mm NMR tubes, and the measurements were performed manually, without automation mode, at 300 ± 0.1 K with 4 dummy scans. ^1^H NMR spectra were obtained by using an Ascend 400 MHz Bruker spectrometer (Bruker France SAS, Wissembourg, France) equipped with a 5 mm BBO probe head. The pulse program with water presaturation used was (*zgcppr*) with 128 scans, 6.83 s acquisition time, 2 s relaxation delay, 64 K data points for fid size, and 4795.396 Hz spectral width. All spectra were manually phased and baseline-corrected and referenced to a TSP signal at 0 ppm. Because the focus is on the minor components of wine, the water signal was suppressed with a power level of 33 W, this value being considered optimum after performing several tests.

### 2.3. Determination of Relative Distribution of Deuterium

The D/H_I_ and D/H_II_ isotopic ratios in ethanol extracted from wine were determined following the OIV-MA-AS311-05 method, as outlined by the International Organization of Vine and Wine (OIV). This method involves the determination of deuterium distribution in ethanol derived from the fermentation of grape musts, concentrated grape musts, grape sugar (rectified concentrated grape musts), and wines using nuclear magnetic resonance (SNIF-NMR). Ethanol was obtained with an alcoholic strength of at least 92% *w*/*w* and a yield of 90% using an automated Cadiot column unit (ADCS Eurofins, Nantes, France). The deuterium–hydrogen (D/H) ratios on the methyl group (CH_2_DCH_2_OH, referred to as D/H_I_) and the methylene group (CH_3_CHDOH, referred to as D/H_II_) were measured using an SNIF probe head with a fluorine lock. Each sample underwent 6 replicates based on 200 scans. For quantification, N,N-Tetramethylurea (Institute for Reference Materials and Measurements, Geel, Belgium) was used as the internal standard, and hexafluorobenzene (Sigma-Aldrich, Darmstadt, Germany) was added as a solvent for the lock signal. To ensure quality control, some measurements were conducted using certified reference materials such as ERM-AE200a, ERM-AE200b, and ERM-AE200c provided by the Joint Research Centre, Institute for Reference Materials and Measurements (IRMM), Geel (Belgium).

### 2.4. Determination of Carbon and Oxygen Stable Isotope Ratios

The ^13^C/^12^C isotopic ratio was determined in ethanol obtained in the same manner as the D/H_I_ and D/H_II_ isotopic ratios, following the method specified in OIV-MA-AS312-06: Determination by Isotope Ratio Mass Spectrometry ^13^C/^12^C of Wine Ethanol or Ethanol Obtained Through the Fermentation of Musts, Concentrated Musts, or Grape Sugar (Resolution Oeno 17/2001). For this determination, 0.1 µL of ethanol was combusted in a Flash EA1112 HT elemental analyzer (Thermo Fisher Scientific, Bremen, Germany) and quantified using a Delta-V IRMS. The data were processed using Isodat 2.5 software. The ^18^O/^16^O isotopic ratio was measured in the water extracted from wine according to the method OIV-MA-AS2-12: Method for ^18^O/^16^O Isotope Ratio Determination of Water in Wines and Musts (Resolution OIV-Oeno 353/2009). The δ^18^O was measured using a continuous-flow isotope ratio mass spectrometer (CF-IRMS) Delta V Plus (Thermo, Bremen, Germany), coupled with a GasBench II isotopic equilibration module. The results were processed using Isodat 3.0 software. For each sample, 500 µL of extracted water was equilibrated for 20 h at 24 °C with a gas mixture containing 0.36% CO_2_ in helium. To ensure the validity of the results, working materials laboratory standards IA-R052, IA-R053, and IA-R054 (ISO Analytical Limited, Crewe, UK) were used.

### 2.5. Statistical Analysis

The large dataset of complex metabolomic profiles and stable isotope ratios can be irrelevant or could hide certain important information without applying suitable multivariate statistical analysis. Therefore, the one-way ANOVA was performed to evaluate the difference between the means and same variance between stable isotope ratios and metabolites at the *p* < 0.05 probability level with Tukey’s test. In addition, the same data were subjected to principal component analysis (PCA) and further to discriminant analysis (DA) to highlight the applicability of metabolites and their stable isotope signatures in classifying the red and white wine samples according to grape variety, vintage, and geographical origin, even if the provenience regions are agroclimatically close. Statistical analysis was performed by using Addinsoft XLSTAT software version 2014.5.03 (Addinsoft Inc., New York, NY, USA).

## 3. Results and Discussion

^1^H-NMR spectra were recorded to identify the metabolites in the wine samples, aiming to discriminate them based on variety, geographical origin, and harvest year. Figure 1 shows the typical 400 MHz proton NMR spectra with water suppression. Among the three spectral regions—aliphatic, carbohydrates, and aromatic—a total of 30 metabolites were identified. Their chemical shifts and multiplicities of the signals taken in consideration for quantification are presented in Table 2. Metabolite identification was made by consulting relevant literature studies [12,30,33,35,36,37,38,39,40] and the Biological Magnetic Resonance Data Bank (https://bmrb.io/, accessed on 15 June 2024). The quantified signal intensities were input into Microsoft Excel (Microsoft^®^ Excel^®^ 2016 MSO, version 2409, build 16.0.18025.20160) for further analysis.

It was remarkable that the peak signal with the highest intensity in the typical ^1^H-NMR wine spectrum was for ethanol; if the water had not been suppressed, these would be the main components of wine, but to differentiate the samples, minor components play a pivotal role. Generally, the spectra are dominated by organic acids (succinic acid, acetic acid, glycerol, 2,3-butanediol, lactic acid), amino acids (proline, alanine), carbohydrates (glucose, fructose), and phenols (caftaric acid, shikimic acid). The organic acids are the class of metabolites with the highest contribution to the stability, flavor, color, and balance of wine, these being present in both white and red wines [41]. Malic acid, for example, plays a crucial role in acidity, metabolizing during ripening. Also, acetic, citric, and succinic acids contribute significantly to the acidity and flavor profile of both red and white wines [42]. The amino acids belong to the organic acids class and add to the wine’s organoleptic properties by generating volatile aroma compounds—essential for achieving the desired acidification during winemaking [43,44,45,46,47]. Regarding the sugars, dry wines typically have them at residual levels, while sweet wines contain higher concentrations, because of their high concentration in ripe grape, constituted primarily by almost identical amounts of glucose and fructose and some sucrose. During the must fermentation, the glucose consumption starts immediately and at a faster rate; thus, the fructose proportion increases throughout the process [48]. But, in the final stages of fermentation, the ethanol concentration increases gradually with the low depletion of sugars and other nutrients [49].

A valuable by-product for certain wines that result in abundance at the beginning of yeast alcoholic fermentation is glycerol, which contributes to wine quality by giving textural properties and mouth-feel attributes. Polyphenols contribute to sensory characteristics and antioxidant activity of wines [50]. Among these, caftaric acid is a phenolic derivate present in high concentration in grape seeds and juice resulting from caffeic and tartaric acids [51]. Also, shikimic acid is a precursor for aromatic amino acid phenylalanine and tyrosine and is used to distinguish grape variety as a marker in fraud control, especially for Pinot Noir, Pinot Gris, and Pinot Blanc [52,53].

Figure 2 illustrates the distribution of metabolite classes across four grape varieties (*Cabernet Sauvignon*, *Crâmpoșie*, *Negru de Drăgășani*, and *Sauvignon Blanc*) cultivated in the investigated areas (Spârleni, Drăgășani, Dobrușa, and Sâmburești), since they are the most representative of their respective regions. It is important to note that achieving complete homogeneity across these regions was not possible due to variations in grape cultivation and production focus within each area. Therefore, the samples were chosen to reflect the typical varieties and styles produced in each region, ensuring that the unique characteristics of each terroir were represented.

Looking at the data, the Cabernet Sauvignon and Negru de Drăgășani varieties exhibited the highest concentrations of aliphatic compounds across the majority of regions. Specifically, the variance in aliphatic compounds was more pronounced in the Sâmburești and Dobrușa regions, suggesting greater regional heterogeneity in these two regions. For example, Cabernet Sauvignon in Sâmburești displayed aliphatic compound values ranging from 463.19 to 4575.92 mg/L, indicating a broad compositional spread compared to tighter distributions observed in other regions. Carbohydrate content demonstrated clear varietal and regional trends, with Cabernet Sauvignon exhibiting the highest levels across all regions and Sauvignon Blanc consistently presenting the lowest values. In Drăgășani and Spârleni, the distribution of carbohydrates was notably more constrained, particularly for Negru de Drăgășani and Crâmpoșie, where values such as 11,532.79 and 9733.60 mg/L respectively, demonstrated more consistent profiles across regions. For aromatic compounds, Cabernet Sauvignon consistently recorded higher levels compared to other varieties across all regions, with Sâmburești showing the most significant elevation, where Cabernet Sauvignon reached up to 463.19 mg/L. Conversely, Crâmpoșie and Sauvignon Blanc exhibited lower and more uniform levels of aromatic compounds, with tighter distributions across all regions, as evidenced by values such as 154.75 mg/L for Crâmpoșie in Drăgășani and 119.65 mg/L for Sauvignon Blanc. The observed variability in aliphatic compounds, carbohydrates, and aromatic compounds across regions and varieties underscores the complex interplay between genetic and environmental factors, which ultimately shape the organoleptic properties of the wines produced in these regions.

The concentration of organic acids and volatile compounds in the wines was lower than of carbohydrates but significantly higher than that of phenols, varying between 1617 mg/L (Sauvignon Blanc, Drăgășani) to 6258 mg/L (Negru de Drăgășani, Sâmburești). The carbohydrate content in wines ranged between 8285 mg/L (Crâmpoșie, Drăgășani) and 14,662 mg/L (Cabernet Sauvignon, Drăgășani), with an inverse relationship observed between carbohydrate content and the levels of organic acids and volatiles. The phenol content was very small compared to the rest of the organic compounds, ranging from 23 mg/L (Sauvignon Blanc, Drăgășani) to 583 mg/L (Cabernet Sauvignon, Spârleni), and the lowest values were found in white wines. These results are in agreement with the reports of other authors who examined the chemical composition of wines produced in different regions from various grape varieties [54,55,56]. Prior to must fermentation, the glucose and fructose concentrations are equal, but during this process, the glucose is preferentially converted by the yeast into alcohol and carbon dioxide, resulting in a higher concentration of fructose in wines [55].

Regarding the stable isotope fingerprints shown in Figure 3, it is notable that the largest range of δ^13^C values was observed in Cabernet Sauvignon, with the most depleted values recorded for samples from Spârleni and the most enriched for a sample from Sâmburești; these are the most geographically distant regions, presenting a difference of 0.9‰ between the mean values. Similarly, the most depleted δ^18^O value was found in Cabernet Sauvignon from Dobrușa, while the most enriched one was observed in Negru de Drăgășani from Sâmburești. For deuterium, the lowest values for both the methyl and methylene groups in ethanol were detected in Cabernet Sauvignon from Spârleni, with the highest values found in Negru de Drăgășani from Drăgășani. These extreme isotope values were observed exclusively in red wines. To enhance the clarity and interpretability of these findings, the data presented in Figure 3 were normalized. Normalized data often result in clearer and more interpretable visualizations, making trends and patterns more apparent. In this context, normalization allowed for a more straightforward comparison of the stable isotope variations across different wines and regions, highlighting the distinct geographical and varietal signatures with greater clarity. The observed trends suggest that grape variety plays a significant role in how isotopic signatures are expressed, reflecting both genetic and environmental factors. Cabernet Sauvignon consistently shows higher isotope values across all categories, potentially indicating that it grows in regions with more stress factors (e.g., drought or evaporation) or has particular physiological traits that enhance its isotopic signature. Negru de Drăgășani and Sauvignon Blanc varieties also display distinct isotope patterns, which might be related to their adaptation to specific climates or differences in water uptake efficiency.

For a closer examination of metabolites variance, specifically organic acids, carbohydrates, and phenols, across different wine varieties, ANOVA was applied. The color gradient applied follows a heatmap convention: red represents higher concentrations, while green represents lower concentrations, with yellow and orange indicating intermediate values. Letters next to the values indicate statistical significance groups based on a Tukey’s post hoc test. Values sharing the same letter within a column are not significantly different (*p* > 0.05), while values with different letters are statistically distinct.

Glycerol was identified as the most abundant metabolite in all wine samples, which is expected, given its normal concentration in dry wine as a by-product of alcoholic fermentation, followed by glucose, 2,3-butanediol, succinic acid, and fructose, while trigonelline, tyrosine, shikimic acid, sucrose, galactose, fumaric acid, formic acid, choline, GABA, alanine, and acetoin were present in lower quantities (Table 3). Glycerol production plays a crucial role in yeast metabolism, contributing to osmotic stress elimination and keeping the oxidation–reduction balance (https://www.decanter.com/learn/what-is-glycerol-ask-decanter-445840/, accessed on 15 June 2024). Also, 2,3-butanediol is a by-product fermentation derived from pyruvic acid, as acetoin, which exhibits neutral organoleptic properties [2]. Notable differences were in the case of malic and citric acids, which were undetectable in certain red wines, such as Merlot, Fetească Neagră, and Negru de Drăgășani, yet were present in high concentrations in white wines, in contrast to lactic acid. This finding can be explained by the fact that the malic acid is converted to lactic acid during malolactic fermentation, especially in red wines [55,57,58].

Regarding the harvest year (Table 4), the ANOVA statistical model revealed the same variance tendency for proline and galacturonic acid. Proline is an important amino acid, and its concentration in plants is strongly influenced by climatic conditions having an osmoprotective role. Also, galacturonic acid concentration increases in plants as a response to long-term drought stress, having an important role in oxidative damage [59].

In regard to regions (Table 5), there can be observed variance of sorbic acid, acetic acid, methanol, citric acid, glucose, galactose and galacturonic acid. From these, the most visible is for galacturonic acid which was also found in other studies as a potential marker for region due to the influence of grape origin on the glycosyl residue content from the wine oligosaccharides [60,61].

Building on the metabolite variance analysis, we further classified all wine samples according to harvest year, grape variety, and geographical origin using principal component analysis (PCA) and discriminant analysis (DA). This classification was crucial in understanding how these factors influence the chemical composition of the wines. For harvest year differentiation, a PCA model based on the identified and quantified metabolites and stable isotope ratios from all wine samples was generated. In the case of metabolites, the contribution from the total variance for the first 5 components, F1, F2, F3, F4, and F5, were 39.77%, 10.41%, 8.86%, 6.06%, and 5.44%, respectively. Figure 4 illustrates the PCA score plot derived from the first two principal components, which together explained 50.18% of the total variance in metabolites and 76.30% of the total variance in stable isotope ratios.

Analysis of the PCA score plot reveals a slight separation between wines produced in 2019 and those from 2020, primarily driven by variations in alanine and fructose, which are positively correlated with the second principal component (F2). This separation suggests that the metabolic profile of these wines is influenced by factors that vary between these two harvest years. In contrast, wines from the 2021 and 2022 harvests show only a tendency toward separation, indicating less pronounced differences in their metabolite compositions. The PCA based on stable isotope ratios provided a more distinct result, with a notable separation observed between the 2020 and 2021 vintages. This separation is largely attributed to variations in δ^18^O ratios, which are known to be strongly influenced by precipitation patterns. The significant differences in precipitation between these two years likely contributed to the observed isotopic distinctions, highlighting the sensitivity of stable isotope ratios to environmental conditions. These findings underscore the importance of considering both metabolite profiles and stable isotope ratios in wine classification, as they can reflect subtle, yet meaningful, differences related to vintage and climatic conditions.

Following the analysis of harvest year variations, Figure 5 illustrates the geographical origins of the wines as influenced by metabolites and stable isotope variations.

As in the case of harvest year, there can be observed a tendency of separation, particularly between wines from Dobrușa and Sâmburești, due to the variation in the signals of glycerol, GABA, and succinic acid, which are positively correlated to F1. Regarding the other two regions, Drăgășani and Spârleni, the PCA analysis did not reveal clear separation, indicating that the metabolic profiles of these regions are more closely aligned. However, the PCA based on stable isotopes revealed some separations between Sâmburești and Spârleni or Dobrușa due to δ^13^C and R relative distribution of deuterium. The samples from Drăgășani appear centrally on the PCA plot, reflecting the region’s geographic position relative to the others. This central placement also suggests that the isotopic composition of wines from Drăgășani, particularly in terms of deuterium and δ^18^O, is influenced by regional climatic factors such as precipitation, which tend to be more consistent across this region.

Building upon the previous analyses, the application of metabolomics allowed for a clear distinction between red and white wines. Figure 6 shows this separation, where Sauvignon Blanc, Cramposie, and Chardonnay wines are distinctly separated from Cabernet Sauvignon, Feteasca Neagra, Merlot, and Negru de Dragasani due to the variance in the malic acid and glucose, which were negatively correlated with F1 and galactose, galacturonic acid, choline, succinic acid, proline, and tyrosine, which were positively correlated with F1. In contrast, when analyzing stable isotope data, the samples did not exhibit a clear separation, suggesting that metabolomic profiling provides a more distinct grouping of wines by variety.

To further validate the findings from principal component analysis (PCA) and uncover additional correlations, the data set was subjected to discriminant analysis (DA). While PCA is an unsupervised technique commonly used to identify patterns and reduce dimensionality, leading to the discovery of distinct separations, DA is a supervised method designed to maximize the separation between predefined groups. In this case, DA provided a higher level of discrimination compared to PCA, particularly when assessing the wines by geographical origin. As shown in Figure 7, the first two discriminant functions (F1 and F2) explained 98.37% of the total variance, with 90.76% attributed to F1 and 7.61% to F2. A significant separation was observed between wines from Dobrușa and those from the other studied regions, driven primarily by the metabolites sorbic acid, arabinose, and tyrosine, which were positively correlated with F1. In addition, shikimic acid, glycerol, isobutanol, and choline further contributed to this differentiation, being positively correlated with F1 as well. It is well documented that the content of fatty acids, amino acids, organic acids, carbohydrates, and phenolic compounds in plants can be influenced by soil nutrient composition [62,63]. For instance, research has shown that nutrient deficiencies in soils can lead to a decrease in alanine, proline, and tyrosine levels in plants [62], which may account for some of the observed metabolite variations in the wines. This association between soil composition and metabolite profiles in wine has been explored in various studies, particularly through the use of NMR-based metabolomics. As highlighted by Bambina et al. [62], metabolomics has been widely used to classify wines according to their geographical origin, but also to variety, vintage, and winemaking techniques. Early research demonstrated that wines from different regions or countries could be clearly differentiated using NMR analysis [27,33]. However, later studies showed that even wines made from the same grape variety, when vinified in different regions, exhibited distinct metabolomic profiles [24,55], suggesting a strong environmental influence, particularly from soil. On a more localized scale, NMR-based metabolomics has also been effective in distinguishing wines within the same geographical area but under different regional trademarks [24,36,46,63,64,65]. For example, Mazzei et al. [24] classified an Italian wine obtained from same varieties of grapes collected from nearby soils with different pedological properties and marked variability in α-hydroxyisobutyrate, lactic acid, succinic acid, glycerol, α-fructose, and β-d-glucuronic acid.

Later, Bambina et al. (2023) [65] revealed the impact of soil chemical–physical parameters on the metabolic profile of wines by showing their correlation with gallic acid, 1-butanol, glycerol proline, threonine, citric acid, succinic acid, shikimic acid, 1-propanol, myo-inositol, choline, acetic acid, and trigonelline. This conclusion was further corroborated by subsequent studies, which continue to affirm the critical role of soil in shaping the chemical composition and distinctiveness of wines [62]. Further supporting this, the metabolites most commonly associated with geographical discrimination include proline, which is known to vary significantly depending on the region, along with alanine, leucine, threonine, and histidine [66,67]. Organic acids such as malic and citric acid, as well as fermentation-derived compounds like lactic acid and succinic acid, have also been found to vary regionally [24,68]. Additionally, metabolites such as 2,3-butanediol, phenethyl alcohol, and glycerol, all products of fermentation, frequently serve as markers of geographical origin [66,67,69]. Among the phenolic compounds, gallic acid has been particularly linked to differences between regions [33,67,70]. These findings reinforce the idea that metabolomics can effectively differentiate wines not only at the country or regional level but also within smaller, geographically close areas. When analyzing stable isotopes, a clear separation can be observed, particularly between Sâmburești and the other regions, due to δ^13^C, which was negatively correlated with F1. Further isotope analysis revealed that separation was more pronounced when considering both δ^13^C and D/H_I_, which are negatively correlated with F1, as well as the relative distribution of deuterium R and D/H_II_, which are positively correlated to F1. The samples from Drăgășani appeared centrally on the graph, reflecting not only the geographical position of this region relative to the others but also the influence of isotopes such as deuterium and δ^18^O, which are strongly affected by local precipitation and climatic conditions. This central placement suggests a more homogeneous isotopic signature for Drăgășani wines, likely due to the moderating environmental factors in the region.

Figure 8 reveals a contribution of 87.59% from the total of metabolite variance for the first two functions of the discriminant analysis, and the differentiation of the wine samples was achieved according to the 4 years of harvest (2019, 2020, 2021, and 2022). F1 has a contribution of 54.28%, followed by F2 with 33.32%, offering a good discrimination for the years 2019, 2020, and 2021.

The metabolites that showed significant variations in the wine samples are sucrose, malic acid, formic acid, myo-inositol, and 1-propanol, which correlated negatively with F1 and sorbic acid, glycerol, GABA, trigonelline, and fructose, with positive coefficient values. A clear separation was observed between the year 2022 and the other years based on the F2 factor. This distinction was influenced by compounds such as shikimic acid, proline, methanol, fumaric acid, galactose, galacturonic acid, and acetoin, which had negative coefficient values, while alanine, succinic acid, and citric acid had positive values. Similar findings have been reported by other researchers. For instance, in a study by Viggiani and Morelli [69], it was noted that proline levels increase with wine age, and succinic acid shows significant variation across wines from different years, making it useful for differentiating vintage. The same authors noted the possibility of using these metabolites for geographical origin discrimination. A different approach focused on differentiating the wines produced in different months revealed higher contents of glycerol, succinic acid, and 2,3-butanediol for products obtained in December, while proline and lactic acid were more concentrated in wines from July, demonstrating the influence of climatic conditions on metabolite content [71]. However, the study realized by Geană et al. [72] indicates a clear discrimination between wines from different years based on alanine and carbohydrates combined with certain anthocyanins and stable isotopes.

From the point of view of the harvest year (Figure 8), based on isotopic composition, in the 2019 samples, there was observed a slight tendency of separation from the 2021 samples due to the variable ^18^O, which is positively correlated with F1.

Regarding the variety (Figure 9), DA analysis revealed a clear separation between red and white wines, based on metabolite data, but there also can be observed a grouping of wines from the same classes, the two first factors summing to 79.11% of the total variance. The metabolites which are contributing to white and red wine separation are citric acid, malic acid, alanine, and glucose, with a negative correlation with F1, and proline, choline, GABA, succinic acid, and tyrosine, with a positive correlation. The separation tendency of white wines (Fetească Regală, Sauvignon Blanc, Crâmpoșie) can be attributed to 2,3-butanediol, galactose, and caftaric and galacturonic acids, which are negatively correlated with F2. Regarding the red wines (Cabernet Sauvignon, Negru de Drăgășani, Fetească Neagră, Merlot), the separation tendency is given by isopentanol and formic and shikimic acids correlated positively with F2.

The obtained results are in agreement with previous studies [23,25,39,63,73,74] focused on wines classification, where the efficiency of ^1^H NMR spectroscopy to be applied in discrimination the variety or in detecting different blends was demonstrated. Godelmann et al. [23] successfully classified different varieties of white and red wines, highlighting the importance of shikimic acid and 2,3-butanediol in prediction of grape variety. Beside these metabolites, also responsible for a high degree of prediction were methanol, lactic acid, citric acid, malic acid, succinic acid, acetic acid, fumaric acid, tartaric acid, 3-methylbutanediol, acetone, and alanine. From the results obtained through application of one- and two-dimensional NMR spectroscopy combined with the PLS-DA model, Papotti et al. [63] revealed a good discrimination based on metabolites present in the low-frequency spectral region, such as 2,3-butanediol, lactic acid, succinic acid, threonine, and malic acid.

Stable isotope ratios, while potentially useful in the majority of contexts related to food authenticity, played a similar role in distinguishing between wine samples in this study. Although stable isotope analysis can sometimes aid in varietal classification, its influence in this case was minimal, suggesting that other factors may be more important in distinguishing between wine varieties. The variety discrimination is an important issue taking into account that some wines resulting from blended grape varieties.

The results of this study demonstrate the effectiveness of integrating stable isotope analysis with NMR-based metabolomics for differentiating wines from geographically proximate, cross-border regions. Previous studies have primarily utilized multivariate statistical methods based on NMR data to classify wines based on their geographical origin, grape variety, and vintage. For example, over 90% classification accuracy was achieved for German wines, but challenges were encountered in distinguishing wines from adjacent regions where climatic conditions were similar [23]. Also, the separation in terms of variety and region of wines from continental areas of France, California, Australia, and Korea was achieved by using NMR spectroscopic analysis (due to some metabolites such as 2,3-butanediol, lactate, acetate, succinate, malate, glycerol, tartrate, glucose, and proline) [33]. Similarly, studies on Pinot Noir wines from Australia and New Zealand have demonstrated differentiation via ICP-MS combined with PCA, underscoring the efficacy of multivariate analysis for geographical wine classification [75]. Furthermore, the differentiation of wines (Cabernet Sauvignon, Blaufrankisch, Merlot, and Pinot Noir) from two geographical regions from Hungary was ascribed to different climate conditions [70]. From the point of view of variety, their results were in accordance with another study based on Czech wine investigation performed by Mascellani et al. [76]. However, these approaches generally focus on broader geographical distinctions, with limited ability to address the specific challenge posed by cross-border regions that share similar agroclimatic conditions. This pilot study builds upon these existing methods by addressing a central problem in the wine industry: the need for effective differentiation of wines from cross-border areas. The used approach incorporates stable isotope analysis alongside metabolomics to capture subtle, yet significant, variations influenced by environmental factors, even when those factors may be largely similar due to geographical proximity. This combination not only offers a more precise classification based on isotopic signatures (such as δ^13^C, δ^18^O, and deuterium ratios), but also enhances the discrimination potential by integrating metabolite profiles, including organic acids, carbohydrates, and phenols, that are unique to specific terroirs. Future research could expand upon these findings by incorporating additional agroclimatic data and examining a broader range of wine-producing regions to further validate and refine this approach.

## 4. Limitations and Future Directions

This study successfully applied ^1^H-NMR and IRMS techniques to differentiate wines from geographically close regions, contributing valuable insights into wine authentication. However, several limitations should be acknowledged. First, the study was restricted to a limited number of grape varieties and regional samples, which may affect the generalizability of the findings across a broader range of wine-producing areas. Future research should aim to include a more diverse set of wine samples, encompassing various regions and grape varieties, to enhance the robustness and applicability of these methods. Additionally, the study focused on stable isotope and metabolomic profiling, which, while effective, may not capture the full spectrum of regional and varietal characteristics influenced by other environmental factors, such as microclimatic variations and soil composition. Future studies could incorporate complementary analytical techniques, such as elemental analysis or the inclusion of pedoclimatic data, to provide a more comprehensive understanding of the terroir effects on wine composition.

## 5. Conclusions

This study underscores the potential of metabolomic profiling and stable isotope analysis as robust tools for cross-border wine authentication. By analyzing red and white wines from four agroclimatically similar regions, we identified significant differences in metabolite content and isotopic composition that facilitate the differentiation of wines by region, grape variety, and vintage. Malic and citric acids were particularly valuable in distinguishing white wines, as their absence in red wines serves as a marker to detect potential blending or dilution with white varieties. Principal component analysis (PCA) revealed patterns related to vintage and region, while discriminant analysis (DA) provided clearer distinctions based on variety, with metabolites such as 2,3-butanediol and shikimic acid being particularly informative. The integration of stable isotope analysis with metabolomics offers a potential approach for verifying the geographic origin of wines from closely situated regions, where climatic similarities may hinder traditional differentiation methods. This cross-border authentication is especially important for wine producers and regulatory authorities aiming to protect the integrity of region-specific wine profiles. By establishing distinct isotopic and chemical signatures, this study demonstrates the feasibility of accurately authenticating wines from neighboring areas, which is crucial for preventing mislabeling and safeguarding regional wine identities. These findings not only enhance our understanding of regional wine characteristics but also lay the foundation for developing a comprehensive database or predictive model to support consistent and reliable authentication practices. For wine producers, these techniques help maintain brand reputation and market differentiation, reinforcing the value associated with Protected Designation of Origin (PDO) and Protected Geographical Indication (PGI) labels. For regulatory bodies, they provide a practical solution for detecting and mitigating cross-border wine fraud, ensuring compliance with origin-related claims, and promoting consumer confidence in the accuracy of wine labeling on an international scale.

## Figures and Tables

**Figure 1 foods-13-03372-f001:**
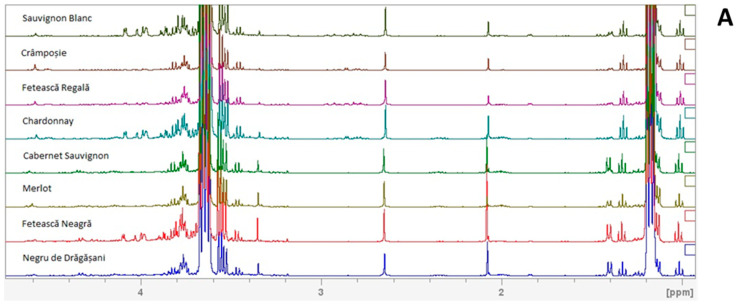
Typical 400 MHz ^1^H-NMR spectra from each sampled variety of wine (Sauvignon Blanc, Crâmpoșie, Fetească Regală, Chardonnay, Cabernet Sauvignon, Merlot, Fetească Neagră, and Negru de Drăgășani): (**A**) aliphatic and carbohydrate regions and (**B**) aromatic region.

**Figure 2 foods-13-03372-f002:**
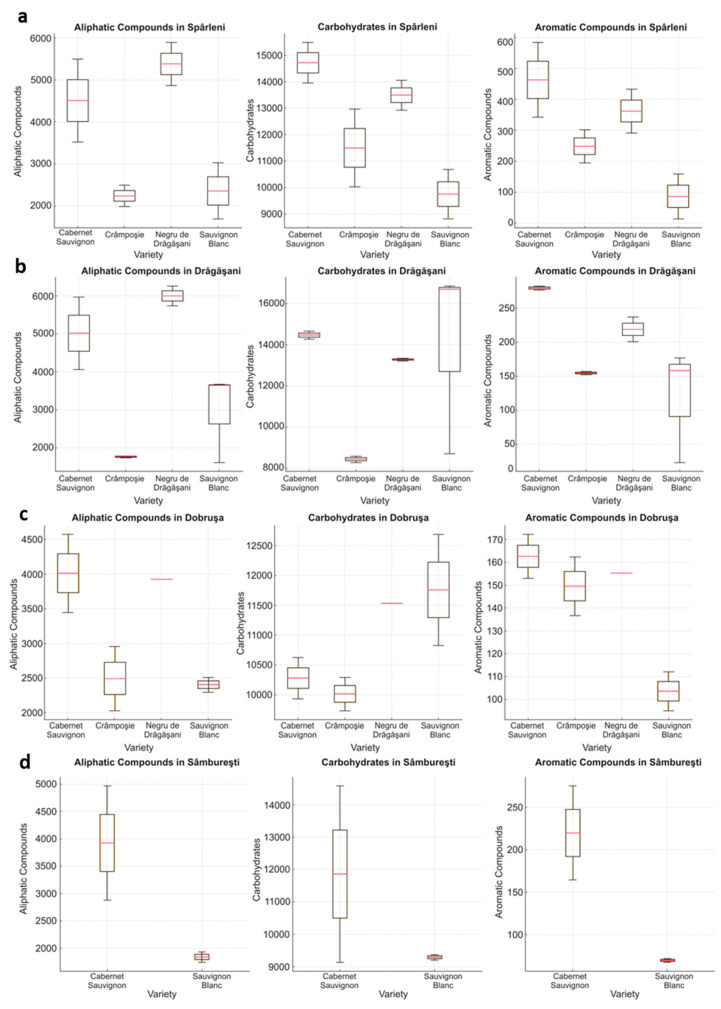
Distribution of wine compositional data (aliphatic compounds, carbohydrates, and aromatic compounds) for the selected varieties (Cabernet Sauvignon, Crâmpoșie, Negru de Drăgășani, and Sauvignon Blanc) across the (**a**) Spârleni, (**b**) Drăgășani, (**c**) Dobrușa, and (**d**) Sâmburești regions.

**Figure 3 foods-13-03372-f003:**
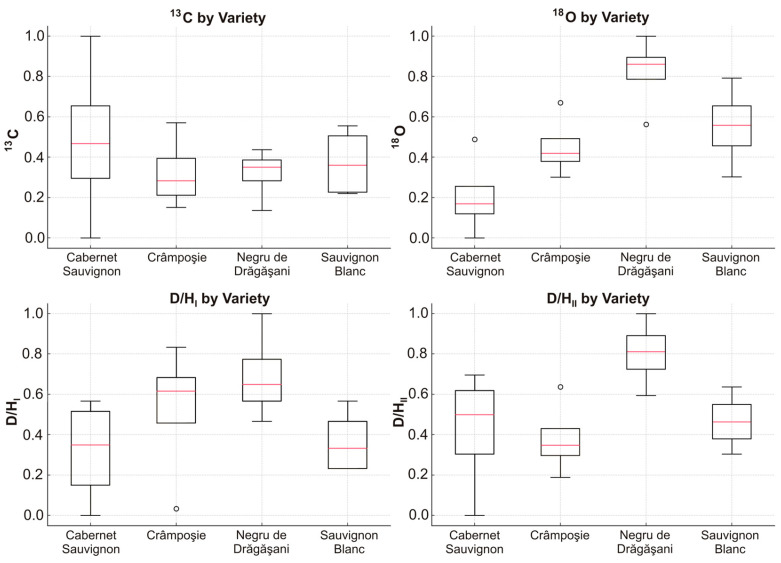
Distribution of isotope normalized values (δ^13^C, δ^18^O, D/H_I_, and D/H_II_), for the selected varieties (Cabernet Sauvignon, Crâmpoșie, Negru de Drăgășani, and Sauvignon Blanc)—the spread, median values, and any potential outliers are visualized.

**Figure 4 foods-13-03372-f004:**
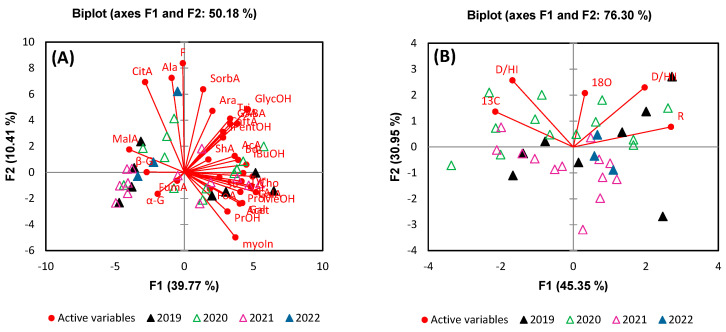
PCA comparing analysis based on (**A**) metabolite content and (**B**) stable isotope ratios for the differentiation of wine samples according to the harvest year.

**Figure 5 foods-13-03372-f005:**
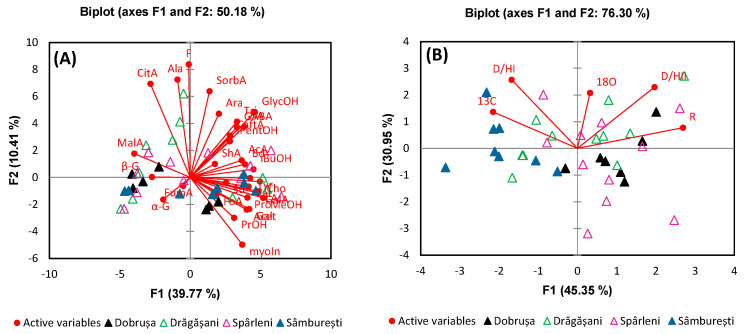
PCA comparing analysis based on (**A**) metabolite content and (**B**) stable isotope ratios for the differentiation of wine samples according to geographical origin.

**Figure 6 foods-13-03372-f006:**
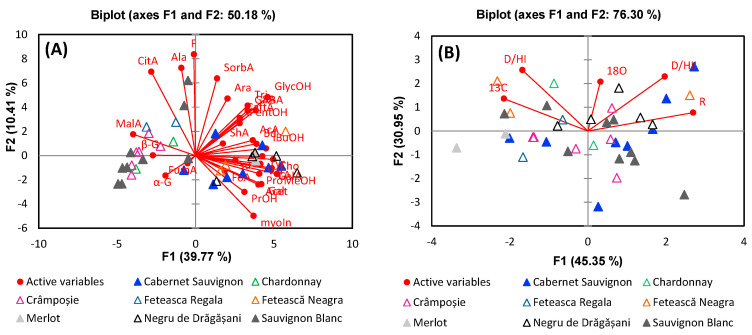
PCA comparing analysis based on (**A**) metabolite content and (**B**) stable isotope ratios for the differentiation of wine samples according to grape variety.

**Figure 7 foods-13-03372-f007:**
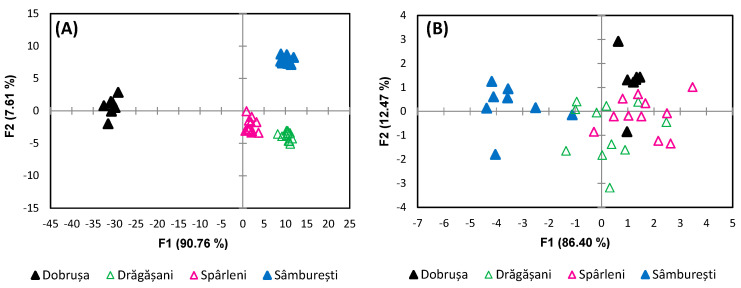
Discrimination of wines’ geographical origin based on (**A**) metabolites and (**B**) stable isotope ratios.

**Figure 8 foods-13-03372-f008:**
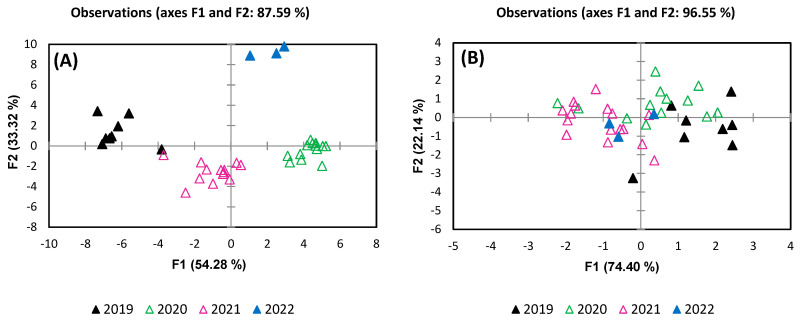
Discrimination of wines’ harvest year based on (**A**) metabolites and (**B**) stable isotope ratios.

**Figure 9 foods-13-03372-f009:**
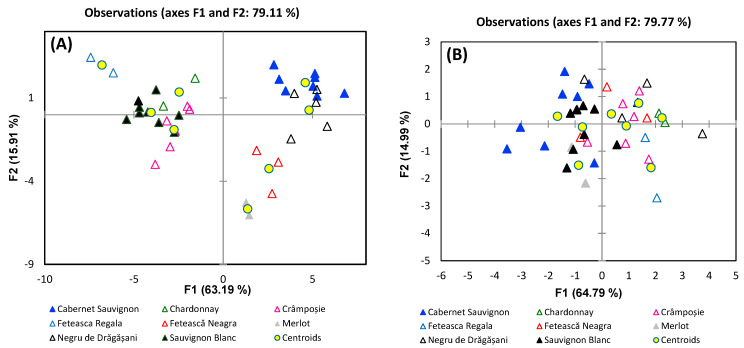
Discrimination of grape variety based on (**A**) metabolites and (**B**) stable isotope ratios.

**Table 1 foods-13-03372-t001:** The wine samples considered for this study.

Crt. No.	Grape Variety	Label Quality Indicator	Type	Colour	Site	Vintage
1	Cabernet Sauvignon	PDO	Dry	red	Dobrușa	2019
2	Negru de Drăgășani	PDO	Dry	red		2020
3	Cabernet Sauvignon	PDO	Dry	red		2021
4	Sauvignon Blanc	PDO	Semi-Dry	white		2021
5	Crâmpoșie	PDO	Dry	white		2021
6	Sauvignon Blanc	PDO	Semi-Dry	white		2022
7	Crâmpoșie	PDO	Dry	white		2022
8	Sauvignon Blanc	PDO	Semi-Dry	white	Drăgășani	2022
9	Negru de Drăgășani	PDO	Dry	red		2020
10	Cabernet Sauvignon	PDO	Dry	red		2019
11	Crâmpoșie	PDO	Dry	white		2021
12	Sauvignon Blanc	PDO	Semi-Dry	white		2021
13	Negru de Drăgășani	PDO	Dry	red		2019
14	Crâmpoșie	PDO	Dry	white		2019
15	Fetească Regală	PDO	Dry	white		2019
16	Fetească Regală	PDO	Dry	white		2020
17	Sauvignon Blanc	PDO	Semi-Dry	white		2020
18	Cabernet Sauvignon	PDO	Dry	red		2021
19	Merlot	PDO	Semi-Dry	red	Sâmburești	2020
20	Fetească Neagră	PDO	Dry	red		2020
21	Sauvignon Blanc	PDO	Semi-Dry	white		2020
22	Cabernet Sauvignon	PDO	Dry	red		2020
23	Cabernet Sauvignon	PDO	Dry	red		2021
24	Sauvignon Blanc	PDO	Semi-Dry	white		2021
25	Fetească Neagră	PDO	Dry	red		2021
26	Merlot	PDO	Semi-Dry	red		2021
27	Merlot	PDO	Semi-Dry	red		2021
28	Cabernet Sauvignon	PDO	Dry	red		2021
29	Cabernet Sauvignon	PDO	Dry	red		2021
30	Sauvignon Blanc	PDO	Semi-Dry	white	Spârleni	2019
31	Chardonnay	PDO	Dry	white		2019
32	Negru de Drăgășani	PDO	Dry	red		2019
33	Crâmpoșie	PDO	Dry	white		2020
34	Chardonnay	PDO	Dry	white		2020
35	Negru de Drăgășani	PDO	Dry	red		2020
36	Cabernet Sauvignon	PDO	Dry	red		2020
37	Fetească Neagră	PDO	Dry	red		2020
38	Cabernet Sauvignon	PDO	Dry	red		2021
39	Crâmpoșie	PDO	Dry	white		2021
40	Sauvignon Blanc	PDO	Semi-Dry	white		2021

**Table 2 foods-13-03372-t002:** The chemical shifts and the multiplicity of the signals ^1^H-NMR appropriate to quantified metabolites.

Crt. No.	Metabolite	Abbreviation	Chemical Shift [ppm]	Multiplicity
1	2,3-Butanediol	Bd	1.132	d
2	Lactic acid	LA	1.407	d
3	Alanine	Ala	1.467	d
4	1-Propanol	PrOH	1.540	m
5	isoPentanol	iPentOH	1.652	m
6	isoButanol	iBuOH	1.729	m
7	Sorbic acid	SorbA	1.841	d
8	Acetic acid	AcA	2.082	s
9	Acetoin	Acet	2.215	s
10	GABA	GABA	2.269	t
11	Proline	Pro	2.352	m
12	Succinic acid	SA	2.653	s
13	Malic acid	MalA	2.895	dd
14	Citric acid	CitA	2.963	d
15	Choline	Cho	3.189	s
16	Methanol	MeOH	3.350	s
17	Glycerol	GlycOH	3.549	dd
18	Fructose	F	3.980	m
19	myo-Inositol	myoIn	4.046	t
20	Arabinose	Ara	4.497	d
21	Glucose	G	4.606 (β)	d
			5.214 (α)	d
22	Galactose	Gal	5.248	d
23	Galacturonic acid	GalA	5.313	d
24	Sucrose	Su	5.430	d
25	Caftaric acid	CaftA	6.422	d
26	Fumaric acid	FumA	6.628	s
27	Shikimic acid	ShA	6.807	m
28	Tyrosine	Tyr	6.840	d
29	Formic acid	FoA	8.278	s
30	Trigonelline	Tri	9.142	s

Signal multiplicity: s—singlet; d—doublet; dd—doublet of doublets; t—triplet; m—multiplet.

**Table 3 foods-13-03372-t003:** Metabolite concentration mean in wines obtained from different grape varieties.

	Bd	LA	Ala	PrOH	iPentOH	iBuOH	SorbA	AcA		Acet	GABA	Pro	SA	Cho	MeOH	FumA
Merlot	826.1 a	619.7 bc	39.9 a	42.5 a	205.3 a	88.6 a	179.1 a	303 abc		16.4 a	11.7 ab	718.3 a	1035.9 a	32.8 a	191.6 a	1.5 a
Fetească Neagra	810.6 a	923.8 ab	33.6 a	53.4 a	135.7 a	60.6 ab	134.9 a	683.5 a		7.2 bc	9.5 ab	324.2 b	809.7 a	29.5 ab	184.5 a	2.0 a
Negru de Drăgășani	664.6 ab	1300 a	42.4 a	65.8 a	256.8 a	83.8 a	83.6 a	615.7 ab		8.4 b	9.1 ab	559.3 a	1045.3 a	35.8 a	161.7 ab	0.6 a
Cabernet Sauvignon	495.1bc	788.5 b	33.3 a	56.7 a	241.7 a	65.2 ab	74.0 a	400 abc		10.9 ab	12.6 a	597.4 a	978.1 a	34.8 a	139.2 b	1.1 a
Feteasca Regala	548.0 bc	74.2 c	52.8 a	46.9 a	233.0 a	35.9 b	176.7 a	165.1 c		1.2 c	8.3 ab	177.1 b	849.7 a	17.8 bc	26.2 c	2.1 a
Sauvignon Blanc	465.6 bc	198.3 c	46.9 a	41.0 a	202.1 a	41.8 b	79.7 a	241.5 bc		0.9 c	8.1 b	184.2 b	761.1 a	14.7 c	32.9 c	1.1 a
Chardonnay	428.0 c	83.6 c	47.7 a	34.8 a	157.6 a	37.3 b	43.8 a	229.3 bc		0.6 c	6.0 b	285.3 b	647.5 a	14.8 c	45.0 c	0.0 a
Crâmpoșie	343.4 c	209.4 c	48.6 a	37.1 a	178.3 a	29.6 b	47.2 a	187.5 bc		0.4 c	5.5 b	199.4 b	712.8 a	16.2 c	23.1 c	1.4 a
	FoA	MalA	CitA	GlycOH	F	myoIn	Ara	β-G	α-G	Gal	GalA	Su	CaftA	ShA	Tyr	Tri
Merlot	11.1 a	0.0 c	0.0 c	11,579ab	219.2 a	295 bcd	181.1 ab	1083 ab	278.5 a	48.8 ab	473.4 a	55.7 a	98.1 ab	16.0 a	55.6 abc	16.9 ab
Fetească Neagra	11.3 a	0.0 c	0.0 c	12,134 a	893.8 a	351 abc	129.1 bc	566.4 ab	221.3 a	53.2 a	480.5 a	35.1 a	147.5 a	45.1 a	67.9 a	19.2 a
Negru de Drăgășani	16.3 a	0.0 c	0.0 c	11,248 abc	323.5 a	430.3 ab	107.7 bc	301.6 b	103.9 a	41.5 ab	396.3 a	63.5 a	135.7 a	52.1 a	61.3 ab	14.3 ab
Cabernet Sauvignon	12.3 a	433.3 bc	85.2 c	10,707 abc	509.1 a	494.1 a	71.8 cd	538.9 b	161.1 a	34.6 bc	284.3 b	30.2 a	94.1 ab	111.8 a	59.1 ab	16.2 ab
Feteasca Regala	16.8 a	1445 ab	1036.0 a	9450 abc	681.0 a	160.5 cd	296.3 a	1404 ab	66.0 a	10.0 d	45.4 c	33.8 a	92.2 ab	17.5 a	39 abcd	13.3 ab
Sauvignon Blanc	10.6 a	1478 ab	596.7 b	8566 abc	757.0 a	227.1 cd	42.3 cd	1501.3 a	314.3 a	16.5 d	86.6 c	28.8 a	55.2 b	7.1 a	24.7 d	10.3
Chardonny	13.2 a	2181.6a	407.0 bc	8130 bc	651.5 a	297 bcd	99.3 bcd	699.4 ab	208.5 a	22.0 cd	126.3 c	22.7 a	23.5 b	28.1 a	31.7 bcd	18.8 a
Crâmpoșie	10.3 a	1301 ab	432.7 b	7788 c	826.0 a	160.1 d	18.8 d	831.5 ab	267.2 a	15.0 d	57.0 c	21.9 a	121.1 a	26.1 a	27.1 cd	9.8 b

**Table 4 foods-13-03372-t004:** Metabolite concentration mean in wines with different vintage.

	Bd	LA	Ala	PrOH	iPentOH	iBuOH	SorbA	AcA	Acet	GABA	Pro	SA		Cho	MeOH	FumA
2019	512.2 a	616.2 a	39.3 b	58.6 a	222.9 a	61.9 a	52.3 a	374.2 a	3.5 a	6.5 a	343.7 a	838.6 a		25.7 a	91.8 a	0.5 ab
2020	577.2 a	678.8 a	43.2 b	44.8 a	196.8 a	57.7 a	123.4 a	423.7 a	6.8 a	9.8 a	416.2 a	913.5 a		24.8 a	106.0 a	0.8 ab
2021	496.2 a	429.6 a	37.4 b	47.1 a	206.5 a	49.6 a	71.3 a	274.9 a	6.2 a	9.5 a	381.6 a	771.1 a		24.6 a	93.7 a	2.4 a
2022	507.0 a	279.3 a	71.1 a	40.1 a	226.9 a	37.1 a	88.7 a	316.8 a	1.1 a	11.0 a	214.8 a	1022.2 a		20.1 a	36.2 a	0.0 b
	FoA	MalA	CitA	GlycOH	F	myoIn	Ara	β-G	α-G	Gal	GalA	Su	CaftA	ShA	Tyr	Tri
2019	14.9 a	997.6 ab	380.9 a	9412.6 a	444.6 a	393.2 a	82.2 a	668.3 a	162.5 a	27.6 a	185.5 a	60.1 a	87.9 a	23.6 a	50.9 a	11.9 a
2020	11.5 ab	383.2 b	266.1 a	10,645 a	866.3 a	290.9 a	114.3 a	667.7 a	182.3 a	31.4 a	269.3 a	30.7 b	106.4 a	44.1 a	49.8 a	16.0 a
2021	11.2 b	1015 ab	218.7 a	9018.8 a	418.3 a	312.3 a	61.1 a	1188.5 a	288.3 a	29.2 a	223.2 a	24.0 b	90.0 a	64.0 a	37.7 a	12.3 a
2022	13.8 ab	1891.8 a	728.7 a	9837.2 a	996.2 a	226.7 a	69.9 a	972.5 a	198.3 a	12.3 a	74.8 a	30.4 b	89.6 a	14.1 a	25.8 a	15.3 a

**Table 5 foods-13-03372-t005:** Metabolite concentration mean in wines obtained from different areas.

	Bd	LA	Ala	PrOH	iPentOH	iBuOH	SorbA		AcA	Acet	GABA	Pro	SA	Cho	MeOH	FumA
Sâmburești	589.6 a	544.0 a	36.0 a	42.8 a	159.4 b	55.2 a	155.5 a		258.4 a	8.6 a	9.6 a	486.4 a	719.1 a	25.3 a	133.1 a	2.2 a
Spârleni	561.3 a	618.7 a	42.9 a	50.0 a	219.2 ab	53.2 a	57.3 b		429.9 a	4.1 a	9.5 a	308.5 a	884.4 a	25.1 a	100.9 a	0.7 a
Drăgășani	485.8 a	496.9 a	43.4 a	48.8 a	259.3 a	62.8 a	120.4 a		366.4 a	3.8 a	8.4 a	353.3 a	967.7 a	24.9 a	77.2 a	0.3 a
Dobrușa	476.6 a	507.5 a	48.1 a	50.4 a	166.6 b	40.3 a	2.2 b		314.0 a	6.0 a	8.5 a	370.5 a	792.5 a	22.0 a	59.0 a	2.1 a
	FoA	MalA	CitA	GlycOH	F	myoIn	Ara	β-G	α-G	Gal	GalA	Su	CaftA	ShA	Tyr	Tri
Sâmburești	10.2 a	467 a	92.8 b	941 a	266 b	341 a	110 a	108 ab	279 a	40.9 a	330 a	33.2 ab	77.5 a	37.3 a	48.4 a	15.1 a
Spârleni	12.0 a	1039 a	206.6 b	10,328 a	948.3 a	310.8 a	71.0 a	438.0 b	267.5 a	30.5 ab	2725 ab	24.5 b	123.7 a	78.0 a	46.1 a	14.7 a
Drăgășani	13.5 a	824.7 a	587.4 a	10,150 a	790 ab	291 a	111.2 a	923 ab	146.4 a	19.2 b	150.5 b	48.1 a	88.9 a	32.0 a	47.2 a	12.4 a
Dobrușa	13.0 a	1085 a	294 ab	8548 a	286 ab	330.8 a	37.0 a	1255 a	175.4 a	24.2 ab	118.1 b	30.7 ab	80.7 a	17.9 a	29.5 a	12.7 a

## Data Availability

The original contributions presented in the study are included in the article/Appendix A, further inquiries can be directed to the corresponding author.

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
