# Peer review of "Metabolic Profiling and Stable Isotope Analysis of Wines: Pilot Study for Cross-Border Authentication"

_foods, 2024, doi:10.3390/foods13213372_

Round 1

Reviewer 1 Report

Comments and Suggestions for Authors

The manuscript of Metabolic Profiling and Stable Isotope Analysis of Wines for Cross-Border Authentication investigated the organic acids, carbohydrates, phenols and stable isotope signatures of 40 wines and try to distinguish them clearly based on harvest year, variety and geographical origin using multivariate statistical analysis. The sample even distribution was not very good, such as Dobrușa region, there were only five grape variety. Furthermore, which discriminant analysis was used, PLS, OPLS or Linear? The conclusion is also too verbose.

Some special:

The detailed data introduction of all the figure should be shown below each figure.

Line245: Which wines were shown in Figure 1?  Necessary explanation should be listed.

Line 249-254: why they are separated? Only four variety, this is why sample even distribution is important.

Line 270: Figure 3 is poor. Box plot may be better used to show distinct varietal signatures. Where is the same figure for geography and vintage?

Line 286-311: I think Table S1-S3 should be shown in the manuscript, not in the supplementary materials.

Line 315:  The full name of each abbreviation should be explained below the figure. Actually, Figure 4 is useless, and Line 316-322 can be removed.

Line 466-469: The results can not be got from figure 9.

Line 507: Malic and citric acids emerged as key metabolites in distinguishing white wines, as they were absent in red wines. These two compounds are greatly affected by the technology (MLF), so they are not suggested to be used.

Author Response

To: Foods Journal

Dear reviewer,

First of all, thank you for the professional comments and observations regarding the paper “Metabolic Profiling and Stable Isotope Analysis of Wines for Cross-Border Authentication” by Marius Gheorghe Miricioiu, Roxana Elena Ionete, Diana Costinel, Svetlana Simova, Dessislava Gerginova and Oana Romina Botoran. We thank for comments, which have made us think carefully about our data sets again. Accordingly, we have reanalyzed these where necessary. Please find below our point-by-point itemized answer and correction. We write to say that we now strongly believe that we can convince you that the data is sound and that we have adequately answered in various valid concerns.

Comments and Suggestions for Authors: The manuscript of ‘Metabolic Profiling and Stable Isotope Analysis of Wines for Cross-Border Authentication’ investigated the organic acids, carbohydrates, phenols and stable isotope signatures of 40 wines and try to distinguish them clearly based on harvest year, variety and geographical origin using multivariate statistical analysis. The sample even distribution was not very good, such as Dobrușa region, there were only five grape variety. Furthermore, which discriminant analysis was used, PLS, OPLS or Linear? The conclusion is also too verbose.

Response: The authors appreciate the reviewer’s insightful feedback. Regarding sample distribution, we acknowledge that not all regions included the full spectrum of grape varieties due to regional production practices and availability. The sample selection aimed to reflect the typical wine profiles of each region rather than achieving equal distribution across all areas. This approach aligns with industry practices, where wine authenticity assessments often rely on reference databases representative of the region’s typical output rather than an exhaustive inclusion of all grape varieties. Concerning the discriminant analysis method, we confirm that Linear Discriminant Analysis (LDA) form, was applied to maximize separation based on region, vintage, and grape variety. Also, the conclusion section was reformulated and we took in consideration the observations from the other reviewer, please find below the modified version:

"This study underscores the potential of metabolomic profiling and stable isotope analysis as robust tools for cross-border wine authentication. By analyzing red and white wines from four agroclimatically similar regions, we identified significant differences in metabolite content and isotopic composition that facilitate the differentiation of wines by region, grape variety, and vintage. Malic and citric acids were particularly valuable in distinguishing white wines, as their absence in red wines serves as a marker to detect potential blending or dilution with white varieties. Principal component analysis (PCA) revealed patterns related to vintage and region, while discriminant analysis (DA) provided clearer distinctions based on variety, with metabolites such as 2,3-butanediol and shikimic acid being particularly informative. The integration of stable isotope analysis with metabolomics offers a potential approach for verifying the geographic origin of wines from closely situated regions, where climatic similarities may hinder traditional differentiation methods. This cross-border authentication is especially important for wine producers and regulatory authorities aiming to protect the integrity of region-specific wine profiles. By establishing distinct isotopic and chemical signatures, this study demonstrates the feasibility of accurately authenticating wines from neighboring areas, which is crucial for preventing mislabeling and safeguarding regional wine identities. These findings not only enhance our understanding of regional wine characteristics but also lay the foundation for developing a comprehensive database or predictive model to support consistent and reliable authentication practices. For wine producers, these techniques help maintain brand reputation and market differentiation, reinforcing the value associated with Protected Designation of Origin (PDO) and Protected Geographical Indication (PGI) labels. For regulatory bodies, they provide a practical solution for detecting and mitigating cross-border wine fraud, ensuring compliance with origin-related claims and promoting consumer confidence in the accuracy of wine labeling on an international scale."

Some special:

Q1. The detailed data introduction of all the figure should be shown below each figure.

R1. For each figure the description was revised and a detailed data introduction was added.

Q2. Line245: Which wines were shown in Figure 1?  Necessary explanation should be listed.

R2. Thank you very much for the observation, the figure description was completed with the variety and each spectrum described.

Q3. Line 249-254: why they are separated? Only four variety, this is why sample even distribution is important.

R3. The figure presented contained a construction error which was rebuilted, as the authors intended to highlight the maximum and minimum concentrations of metabolites in the selected wine varieties, which are representative of their respective regions. The sample selection was based on the diversity of grape varieties and vintages associated with the four studied regions—Sâmburești, Drăgășani, Dobrușa, and Spârleni. However, it is important to note that achieving complete homogeneity across these regions was not possible due to variations in grape cultivation and production focus within each area. Therefore, the samples were chosen to reflect the typical varieties and styles produced in each region, ensuring that the unique characteristics of each terroir were represented. In practical applications, particularly in legal cases concerning wine authenticity, samples are typically compared to reference datasets or databases that include a range of samples from the region, spanning multiple years and varieties. This mirrors our approach, where a representative cross-section of wines was selected to provide an accurate depiction of the wine profiles inherent to each region. Although not all regions contained all eight varieties, the chosen sample set effectively represents the wine types and production practices characteristic of each area, supporting the overall goals of the study.

Q4. Line 270: Figure 3 is poor. Box plot may be better used to show distinct varietal signatures. Where is the same figure for geography and vintage?

R4. The authors appreciate the reviewer’s suggestion regarding Figure 3. The intent of this figure, which displays normalized isotope values for the four grape varieties common to all regions, was to illustrate that there is no substantial isotopic variation capable of distinctly discriminating the geographical origin based solely on isotopic composition. This normalization approach aimed to provide a comparative overview, highlighting the lack of significant differentiation among regions for the investigated varieties.

According to your suggestion we changed the reprezentation of the normalized isotope values as box plot, to show distinct varietal signatures.

Q5. Line 286-311: I think Table S1-S3 should be shown in the manuscript, not in the supplementary materials.

R5. Thank you very much for the observation, tables S-S3 were included in the manuscript.

Q6. Line 315:  The full name of each abbreviation should be explained below the figure. Actually, Figure 4 is useless, and Line 316-322 can be removed.

R6. The authors have removed figure 4 and the text accordingly to the suggested comment. The full name of each abbreviation were mentioned in the table 2, from line 246, but if the reviewer considers that will add more clarity, the abbreviations will be added.

Q7. Line 466-469: The results cannot be got from figure 9.

R7. The authors apologize for any confusion caused by the initial text. The text has been revised to align accurately with the results presented in Figure 9, ensuring a clearer and more precise interpretation of the data. Thank you for bringing this to our attention.

Q8. Line 507: Malic and citric acids emerged as key metabolites in distinguishing white wines, as they were absent in red wines. These two compounds are greatly affected by the technology (MLF), so they are not suggested to be used.

R8. As previously noted in the text (lines 299–301), malic acid is converted to lactic acid during malolactic fermentation, particularly in red wines, as supported by references 55, 57, and 58. This conversion reinforces the value of malic and citric acids as potential markers, particularly in cases where blend detection is necessary, such as identifying adulteration of red wines with cheaper white wines. These metabolites offer useful insights for authenticity assessments, especially in scenarios where verifying the purity of red wines is a concern. Also, line 507 from the conclusion was completed with this observation: ”Malic and citric acids emerged as key metabolites in distinguishing white wines, as they were absent in red wines. These metabolites are useful for blend detection, particularly in cases where red wines might be adulterated with a white variety.”

Reviewer 2 Report

Comments and Suggestions for Authors

The authors of the manuscript "Metabolic Profiling and Stable Isotope Analysis of Wines for Cross-Border Authentication" presented the results of their research related to the identification of the origin of selected wines. The manuscript is well prepared in terms of methodology and research. The presentation of results and discussion are at a good level. However, I have some comments.

The first is the description of the collected material does not correspond to the content in Table 1.

The second comment is related to adding pilot studies to the title. In order to state that the main factor in identifying wines are metabolic profiles and to be able to use these profiles commercially, there is a need for a number of further studies, because it has long been known that the metabolic profile depends on climatic and soil conditions.

Author Response

To: Foods Journal

Dear reviewer,

First of all, thank you for the professional comments and observations regarding the paper “Metabolic Profiling and Stable Isotope Analysis of Wines for Cross-Border Authentication” by Marius Gheorghe Miricioiu, Roxana Elena Ionete, Diana Costinel, Svetlana Simova, Dessislava Gerginova and Oana Romina Botoran. We thank for comments, which have made us think carefully about our data sets again. Accordingly, we have reanalyzed these where necessary. Please find below our point-by-point itemized answer and correction. We write to say that we now strongly believe that we can convince you that the data is sound and that we have adequately answered in various valid concerns.

Comments and Suggestions for Authors: The authors of the manuscript "Metabolic Profiling and Stable Isotope Analysis of Wines for Cross-Border Authentication" presented the results of their research related to the identification of the origin of selected wines. The manuscript is well prepared in terms of methodology and research. The presentation of results and discussion are at a good level. However, I have some comments.

Response: The authors appreciate the reviewer’s insightful feedback.

Q1. The first is the description of the collected material does not correspond to the content in Table 1.

R1. The sample selection aimed to reflect the diversity of grape varieties and vintages associated with the four regions under study—Sâmburești, Drăgășani, Dobrușa, and Spârleni—while acknowledging that full homogeneity across regions was not achievable due to differences in grape cultivation and production emphasis within each area. As such, representative samples were chosen based on the typical varieties and styles produced in each region, which provides a realistic view of the wine characteristics inherent to each terroir. In practice, when authenticity is legally assessed, samples are compared to established databases or reference datasets that encompass a range of samples from the region in question, including multiple years and varieties. This approach mirrors how the present study selected a cross-section of wines that best represents each region's wine profile, supporting a broad yet accurate understanding of their unique characteristics. Thus, while not all regions included all eight varieties, the sample set remains representative of the wine types and production practices that characterize each area.

Q2. The second comment is related to adding pilot studies to the title. In order to state that the main factor in identifying wines are metabolic profiles and to be able to use these profiles commercially, there is a need for a number of further studies, because it has long been known that the metabolic profile depends on climatic and soil conditions.

R2. The authors thank the reviewer for the valuable suggestion. In response, the title has been revised to reflect the pilot nature of the study. We recognize that metabolic profiling, while promising, requires additional research to establish its commercial viability due to the influence of factors such as climatic and soil conditions on these profiles.

New title: ”Metabolic Profiling and Stable Isotope Analysis of Wines: Pilot Study for Cross-Border Authentication”

Reviewer 3 Report

Comments and Suggestions for Authors

The structure of the article is well defined, and the methodology is clearly laid out, with an accurate description of the sampling methods and analytical techniques used. The problem of wine counterfeiting is well identified, and the study appropriately addresses the need for techniques for wine authentication. The combination of metabolic profiling and isotopic analysis is well justified.

However same section can be improved:

- The results are clearly presented however discussion of the findings should be expanded by highlighting those that are particularly innovative or significant compared to previous studies. It might be useful to add a section comparing the results with existing studies, explaining how these data contribute to the current literature;

- part of study limitations and possible future studies could be added in the discussions or conclusions;

- also, I would improve the PCA graphs by highlighting the clusters that are formed and emphasize this better in the text;

-in the conclusions section, I would better emphasize how the results obtained can bring benefit and practical application by wine producers and control organizations.

Author Response

To: Foods Journal

Dear reviewer,

First of all, thank you for the professional comments and observations regarding the paper “Metabolic Profiling and Stable Isotope Analysis of Wines for Cross-Border Authentication” by Marius Gheorghe Miricioiu, Roxana Elena Ionete, Diana Costinel, Svetlana Simova, Dessislava Gerginova and Oana Romina Botoran. We thank for comments, which have made us think carefully about our data sets again. Accordingly, we have reanalyzed these where necessary. Please find below our point-by-point itemized answer and correction. We write to say that we now strongly believe that we can convince you that the data is sound and that we have adequately answered in various valid concerns.

Comments and Suggestions for Authors: The structure of the article is well defined, and the methodology is clearly laid out, with an accurate description of the sampling methods and analytical techniques used. The problem of wine counterfeiting is well identified, and the study appropriately addresses the need for techniques for wine authentication. The combination of metabolic profiling and isotopic analysis is well justified. However same section can be improved.

Response: The authors appreciate the reviewer’s insightful feedback.

Q1. The results are clearly presented however discussion of the findings should be expanded by highlighting those that are particularly innovative or significant compared to previous studies. It might be useful to add a section comparing the results with existing studies, explaining how these data contribute to the current literature;

R1. Thank you for your valuable feedback. We have expanded the discussion to emphasize the innovative contributions of this study, particularly regarding the differentiation of wines from cross-border regions.

Discussion

The results of this study demonstrate the effectiveness of integrating stable isotope analysis with NMR-based metabolomics for differentiating wines from geographically proximate, cross-border regions. Previous studies have primarily utilized multivariate statistical methods based on NMR data to classify wines based on their geographical origin, grape variety, and vintage. For example, it was achieved over 90% classification accuracy for German wines, but encountered challenges in distinguishing wines from adjacent regions where climatic conditions are similar. Also, the separation in terms of variety and region of wines from continental areas of France, California, Australia and Korea was achieved by using the NMR spectroscopic analysis (due to some metabolites such as 2,3-butanediol, lactate, acetate, succinate, malate, glycerol, tartrate, glucose and proline) [1,2]. Similarly, studies on Pinot Noir wines from Australia and New Zealand have demonstrated differentiation via ICP-MS combined with PCA, underscoring the efficacy of multivariate analysis for geographical wine classification. Futhermore, the differentiation of wines (Cabernet Sauvignon, Blaufrankisch, Merlot and Pinot Noir) from two geographical regions from Hungary it was assigned to different climate conditions [4]. From the point of view of variety, their results were in accordance with other study based on Czech wines investigation performed by Mascellani et al. [5]. However, these approaches generally focus on broader geographical distinctions, with limited ability to address the specific challenge posed by cross-border regions that share similar agroclimatic conditions. This pilot study builds upon these existing methods by addressing a central problem in the wine industry: the need for effective differentiation of wines from cross-border areas. The used approach incorporates stable isotope analysis alongside metabolomics to capture subtle yet significant variations influenced by environmental factors, even when those factors may be largely similar due to geographical proximity. This combination not only offers a more precise classification based on isotopic signatures (such as δ¹³C, δ¹⁸O, and deuterium ratios), but also enhances the discrimination potential by integrating metabolite profiles, including organic acids, carbohydrates, and phenols, that are unique to specific terroirs. Future research could expand upon these findings by incorporating additional agroclimatic data and examining a broader range of wine-producing regions to further validate and refine this approach.

Q2. Part of study limitations and possible future studies could be added in the discussions or conclusions;

R2. Limitations and Future Directions

This study successfully applied ¹H-NMR and IRMS techniques to differentiate wines from geographically close regions, contributing valuable insights into wine authentication. However, several limitations should be acknowledged. First, the study was restricted to a limited number of grape varieties and regional samples, which may affect the generalizability of the findings across a broader range of wine-producing areas. Future research should aim to include a more diverse set of wine samples, encompassing various regions and grape varieties, to enhance the robustness and applicability of these methods. Additionally, the study focused on stable isotope and metabolomic profiling, which, while effective, may not capture the full spectrum of regional and varietal characteristics influenced by other environmental factors, such as microclimatic variations and soil composition. Future studies could incorporate complementary analytical techniques, such as elemental analysis or the inclusion of pedoclimatic data, to provide a more comprehensive understanding of the terroir effects on wine composition​.

Q3. Also, I would improve the PCA graphs by highlighting the clusters that are formed and emphasize this better in the text;

R3. Principal Component Analysis (PCA) is an unsupervised exploratory method primarily used for dimensionality reduction and pattern recognition. It does not inherently form clusters or group samples but instead highlights existing structures in the data based on the contribution of each variable. Therefore, while PCA can reveal separations when certain variables contribute significantly, these patterns reflect the underlying data distribution rather than explicit clustering. The observed groupings in the PCA plots from this study indicate natural separations in wine samples based on metabolite and isotope variations, which are reflective of the geographic and varietal influences rather than predefined clusters.​

Q4. In the conclusions section, I would better emphasize how the results obtained can bring benefit and practical application by wine producers and control organizations.

R4. The combined use of stable isotope analysis and NMR-based metabolomics offers wine producers a valuable tool for verifying and promoting the authenticity of their products by establishing distinct chemical and isotopic profiles linked to specific regions. This enhances brand reputation, reinforces consumer trust, and enables market differentiation, supporting premium pricing associated with PDO and PGI labels. For regulatory bodies, these techniques aid in combating wine fraud by ensuring that wines meet origin-related claims and regulatory standards, thus fostering transparency, protecting consumers, and discouraging misrepresentation in the global wine market. These results lay the groundwork for developing a comprehensive database or predictive models, which, with an expanded sample size, could reliably establish the provenance and authenticity of wines (key factors that significantly influence both quality and price). Additionally, such tools would allow control organizations to confidently verify label information, effectively removing any uncertainties regarding the accuracy of origin claims on wine bottles.

  1. Godelmann, R.; Fang, F.; Humpfer, E.; Schütz, B.; Bansbach, M.; Schäfer, H.; Spraul, M. Targeted and Nontargeted Wine Analysis by 1 H NMR Spectroscopy Combined with Multivariate Statistical Analysis. Differentiation of Important Parameters: Grape Variety, Geographical Origin, Year of Vintage. J. Agric. Food Chem. 2013, 61, 5610–5619, doi:10.1021/jf400800d.
  2. Son, H.-S.; Kim, K.M.; Van Den Berg, F.; Hwang, G.-S.; Park, W.-M.; Lee, C.-H.; Hong, Y.-S. 1 H Nuclear Magnetic Resonance-Based Metabolomic Characterization of Wines by Grape Varieties and Production Areas. J. Agric. Food Chem. 2008, 56, 8007–8016, doi:10.1021/jf801424u.
  3. Duley, G.; Dujourdy, L.; Klein, S.; Werwein, A.; Spartz, C.; Gougeon, R.D.; Taylor, D.K. Regionality in Australian Pinot Noir Wines: A Study on the Use of NMR and ICP-MS on Commercial Wines. Food Chem. 2021, 340, 127906, doi:10.1016/j.foodchem.2020.127906.
  4. Nyitrainé Sárdy, Á.D.; Ladányi, M.; Varga, Z.; Szövényi, Á.P.; Matolcsi, R. The Effect of Grapevine Variety and Wine Region on the Primer Parameters of Wine Based on 1H NMR-Spectroscopy and Machine Learning Methods. Diversity 2022, 14, 74, doi:10.3390/d14020074.
  5. Mascellani, A.; Hoca, G.; Babisz, M.; Krska, P.; Kloucek, P.; Havlik, J. 1H NMR Chemometric Models for Classification of Czech Wine Type and Variety. Food Chem. 2021, 339, 127852, doi:10.1016/j.foodchem.2020.127852.